

# On moist ocean-atmosphere coupling mechanisms

Oksana Guba[1], Arjun Sharma[1], Mark A. Taylor[1], Peter A. Bosler[1], and Erika L. Roesler[1]

[1]Sandia National Laboratories, Albuquerque, NM, USA

**Correspondence:** Oksana Guba (onguba@sandia.gov)

**Abstract.** We investigate mechanisms governing moist energy exchanges at the atmosphere-ocean interface in global Earth system models. The goal of this work is to overcome deficiencies like energy fixers and unphysical thermodynamic formulations and designs that are commonly used in modern models. For example, while the ocean surface evaporation is one of the most significant climatological drivers, its representation in numerical models may not be physically accurate. In particular, existing schemes give an incorrect atmospheric air temperature tendency during evaporation events. To remedy this, starting from first principles, we develop a new mechanism for the ocean-atmosphere moist energy transfers. It utilizes consistent thermodynamics of water species, distributes latent heat of evaporation in a physically plausible way, and avoids reliance on artificial energy fixers. The temperature and water mass tendencies are used to formulate a set of ordinary differential equations (ODEs) representing a simple box model of ocean-air exchange. We investigate the properties of the ODEs representing the proposed mechanism and compare them against those derived from the current designs of the Energy Exascale Earth System Model (E3SM). The proposed simplified box model highlights the advantages of our approach in capturing physically appropriate atmospheric temperature changes during evaporation while conserving energy.

## 1 Introduction

The purpose of this work is to investigate mechanisms of latent heat transfer due to evaporation at the ocean-atmosphere interface in climate models. Alongside radiation, the energy fluxes associated with precipitation and evaporation are one of the largest contributors to the Earth climate patterns (Trenberth et al., 2009; Stevens and Schwartz, 2012). A recently published overview (Lauritzen et al., 2022) highlights major deficiencies in the thermodynamic formulations used in the numerical climate models. One of the most significant issues in the models is incorrect representation of the internal energy of water forms in the atmosphere, which leads to errors in the energy footprint of evaporation and precipitation at the atmosphere-ocean interface.

There has been recent research into modeling consistent unapproximated thermodynamics for both the atmospheric (Eldred et al., 2022; Guba et al., 2024) and the ocean (Mayer et al., 2017) components of the models. Unlike many current designs that assign dry heat capacities to all forms of water in the atmosphere, the unapproximate thermodynamics uses close to theoretically established values specific to each water form. Therefore, there are large discrepancies between current designs and designs based on the unapproximated thermodynamics in representing energy fluxes. For example, enthalpy, defined with phase-appropriate specific heat capacities (Vallis, 2017; Eldred et al., 2022), is regarded as a valid representation of the biggest





source of energy fluxes at the lower boundary of the atmosphere (Lauritzen et al., 2022; Guba et al., 2024). Using enthalpy based on unapproximated specific heats of water vapor ($c_p^v = 1870$ J kg$^{-1}$ K$^{-1}$ as defined in Emanuel (1994)) and liquid water ($c_l = 4190$ J kg$^{-1}$ K$^{-1}$), instead of the specific heat of the dry air ($c_p^d = 1005.7$ J kg$^{-1}$ K$^{-1}$), alters the energy signal of water forms by a factor 2 to 4.

Although many of these inconsistencies are patched using global energy fixers and pressure adjustments (Lauritzen and Williamson, 2019; Golaz et al., 2022; Guba et al., 2024), we argue that such approaches mask the underlying problems and limit the fidelity of Earth system models. As model resolution increases and we seek higher accuracy in regional and process-level predictions, continued reliance on artificial fixers becomes increasingly problematic.

In this work, we discuss one of the energy fixers, called IEFLX (Golaz et al., 2022), used in the Earth Exascale Energy System Model (E3SM) (Golaz et al., 2019, 2022) and its atmospheric component, the E3SM Atmosphere Model (EAM) (Rasch et al., 2019). We explain how it restores the energy budget associated with latent heat fluxes from evaporation and precipitation at the atmosphere-ocean interface. While IEFLX balances the energy budget in E3SM, as we show, it does not model latent heat transfers in a physically consistent manner. This deficiency may potentially hinder Earth system models'

fidelity and capabilities as the community transitions to use high-resolution and regional models.

Previously, in Guba et al. (2024), we analyzed precipitation mechanisms with consistent unapproximated thermodynamics. Since there is a delicate balance between climatological energy trends of precipitating and evaporating fluxes at the atmosphere-ocean interface, it is not possible to redesign a numerical climate model gradually, by addressing only one or the other flux first. Instead, improvements in model thermodynamics must be applied to both evaporation and precipitation mechanisms

simultaneously, even though they are often controlled by different components of the model. Therefore, this work, which focuses on both evaporative and precipitating mechanisms relevant to the atmosphere within a framework of unapproximated thermodynamics, is a natural extension of Guba et al. (2024).

Here, we investigate evaporation from the ocean surface as modeled in E3SM. We dive into the details of how the latent heat of evaporation is handled in E3SM with the help of global energy fixers, and how it could be redistributed using the unapprox-

imated thermodynamics without fixers. We argue that the transfer of latent heat from evaporation across the atmosphere-ocean interface is not modeled in a physically plausible manner. To clarify the impact of these formulations, we implement three simplified numerical box models: one using consistent, unapproximated thermodynamics, and two mimicking E3SM-like assumptions. These models describe the temperature and water mass tendencies in the ocean and atmosphere using a system of four coupled ordinary differential equations, representing the evolution of atmospheric and oceanic water mass and temperature

over time. The ocean and atmosphere are each represented as a single, well-mixed box. We show that the model based on consistent thermodynamics produces a different atmospheric temperature tendency during evaporation compared to the E3SM-like models. See Sec. 3.4.5 for details.

The overarching goal of this work is to further investigate deficiencies in thermodynamic approaches in current Earth system models. It aims to direct the Earth system modeling community toward the development of more physically and numerically

consistent models by reducing reliance on crude approximations and artificial fixers. We emphasize that this study does not suggest that using the unapproximate thermodynamics in precipitation and evaporation would affect climatological biases in





the current numerical Earth system models in any particular way. Such biases are often managed through extensive parameter tuning to match observations, and this tuning will likely remain necessary, even with improved physical foundations, for the foreseeable future. Nevertheless, we argue that the advances such as that proposed here can reduce the burden on practitioners

to rely on such ad hoc tuning and enable more interpretable, transparent models grounded in sound physical and mathematical principles.

## 2 Overview and motivation

### 2.1 Moist physics in Earth system models: evaporation and condensation

The motivation for this work is two-fold. First, we aim to raise awareness about crude thermodynamic approximations com-

monly employed in the modern global Earth system models, in particular, in their atmospheric components and at the surface interfaces. Second, we propose conceptual improvements intended to enhance the physical fidelity of these models.

For the purpose of this work, we separate moist physics at the ocean-atmosphere interface into two simplified categories: **condensation** and **evaporation**. For **condensation**, we consider processes that lead to precipitation. In climate models, such processes are typically represented by micro- and macro-physical parametrizations (see, e.g.,Klemp and Wilhelmson (1978);

Morrison and Gettelman (2008); Morrison and Milbrandt (2015); Golaz et al. (2002)). For **evaporation**, we consider only the flux of water vapor from the ocean surface into the atmosphere. Such processes are often modeled by so-called bulk schemes (Haidvogel and Bryan, 1993) based on Monin-Obukhov Similarity Theory (MOST) (Shaw, 1990). Our simplified treatment of condensation and evaporation focuses on the thermodynamics at the ocean–atmosphere interface. In reality—and in more complex model implementations—these processes are not confined neatly to either the ocean or the atmosphere. For example,

a condensed water droplet may remain suspended in the atmosphere or evaporate before reaching the lower boundary of the atmosphere. In our simplified framework, however, we assume that all condensed water mass in the atmosphere is transported directly to the ocean.

The thermodynamic aspects of condensation, along with possible improvements and their implications, were previously discussed by Guba et al. (2024). This work shifts focus to evaporation and the combined effects of evaporation and condensation.

As further discussed in Sec. 2.2.4, while bulk schemes compute mass and temperature fluxes at the atmosphere-ocean interface, they do not account for energy transfers associated with evaporation. Instead, these transfers are modeled separately within the ocean and the atmosphere components of the model. In the following section, we examine the mechanisms governing these evaporative energy transfers in detail, because they provide a clear motivation behind this work.

### 2.2 Motivation: Closer look at energy transfers during evaporation

#### 2.2.1 An overview of definitions and assumptions

In Sec. 3 we will introduce three sets of simple models – two of these are based on the implementation of the ocean and atmosphere thermodynamics in E3SM, and one representing an idealized implementation using unapproximated thermodynamics



in both components. In all models, the ocean and the atmosphere components are represented by mean grid values for species
mass and temperature. Before we get into the details of derivations in Sec. 3, we first motivate for our work by conceptually

examining evaporation at the ocean-atmosphere interface.

Evaporative mechanisms at the ocean-atmosphere interface are incredibly complex (Niiler, 1993; Feistel and Hellmuth,
2023). While evaporation is ultimately driven by solar radiation (Trenberth et al., 2009), the net evaporative flux is influenced
by a combination of external heating, the thermodynamic and dynamic states of both the atmosphere and the ocean, mixing
processes, and even photomolecular effects (Tu et al., 2023).

Here, we focus only on a highly simplified version of one of these mechanisms, namely, the transfer of energy during
evaporation from the ocean surface, in the absence of external heating (i.e., we assume that radiative energy fluxes preceding
evaporation have already been absorbed by the ocean) or dynamical effects (no mixing or surface winds).

In our simplified framework, within the unapproximated thermodynamics, we will use enthalpy to represent internal energy,
consistent with common practive in atmospheric modeling of reducing the conservation of energy to conservation of enthalpy

(Lauritzen et al., 2022; Guba et al., 2024; Yatunin et al., 2025). In the unapproximated case, in both ocean and atmosphere, the
enthalpy of water vapor is  given by (Eldred et al., 2022) $h^v = (c_p^v T + L_v + L_l)m^v$ and that of liquid water by $h^l = (c_l T + L_l)m^l$,
where $T$ is temperature, and $m^v$ and $m^l$ are vapor and liquid water masses. In the current setup, whether we are using specific
or mass-weighted enthalpies will be obvious from the context, and thus, we omit this distinction in the text. Terms including
$L_v = 2.501 \times 10^6$ J kg$^{-1}$ and $L_l = 3.3337 \times 10^5$ J kg$^{-1}$ represent potential energy of molecular bonds.

In many models, like EAM, the atmosphere uses the assumption that heat capacities for water species are the same as for the
dry air, i.e., the enthalpies of vapor and liquid water are given by $h_{approx.}^v = (c_p^d T + L_v + L_l)m^v$ and $h_{approx.}^l = (c_p^d T + L_l)m^l$.

Some older formulations omit the *L terms*, $L_v + L_l$ or $L_l$ from enthalpy definitions. This may lead to confusion when
computing energy exchanges during phase changes. A phase change, for example, from vapor to liquid, can be viewed as a
2-step process: release of latent heat, equal to $h^v - h^l$, and absorption of that energy by the surrounding environment as sensible

heat, thus conserving total energy (or enthalpy). By incorporating the *L terms* directly into the enthalpy definitions, these two
steps are naturally combined into a single energy-conserving computation, as we adopt below in Sections 2.2.2 and 2.2.3.

Another key aspect of evaporation that we emphasize is that the energy of vaporization at the air-water interface must come
from water. While, in reality the process is modulated by large effects of mixing, surface winds, roughness, etc., when these are
neglected as in our simplified setup, evaporation is expected to cool the surface of the ocean surface (Niiler, 1993). This implies

that the energy of vaporization should be drawn from the ocean. In Sec. 2.2.2, we show that E3SM does not fully account for
the energy of vaporization.This shortcoming will be remedied by the new design introduced in Sec. 2.2.3.

### 2.2.2    Current design of E3SM

The ocean component of E3SM is represented by MPAS-Ocean model (Ringler et al., 2013) and, as mentioned above, the
atmosphere is represented by EAM (Rasch et al., 2019). Several options for the surface flux exchange at the atmosphere-ocean

interface are based on the Monin–Obukhov Similarity Theory (MOST) theory, and thus produce water vapor fluxes from the




ocean surface. However, as discussed above and shown in detail below, these schemes do not properly calculate temperature tendencies resulting from the liquid-to-vapor phase transition.

It is common in Earth system models for energy and mass fluxes to be computed independently within each model component. As these model components may use different thermodynamic assumptions, the energy fluxes derived from mass and temperature, necessitating the use of energy fixers, like IEFLX (Golaz et al., 2019) to maintain global energy conservation.

Assume ocean has temperature $T_o$ and total liquid water mass is $m_o^l + \Delta m^l$, where $\Delta m^l$ is the amount of water to be evaporated from the ocean surface (computed using a bulk scheme; see Sec. 2.2.4). In MPAS-Ocean, the energy (enthalpy) of this water is defined as:

$$E_{ocean} = c_l T_o(m_o^l + \Delta m) + L_l(m_o^l + \Delta m). \tag{1}$$

When this mass $\Delta m^l$ is transferred to the atmosphere, it is associated with an energy flux

$$F = c_p^d T_a \Delta m + (L_v + L_l)\Delta m, \tag{2}$$

where $T_a$ is the atmospheric temperature. Note that in the atmospheric component of E3SM, EAM, this energy flux is not received explicitly. The $c_p^d$ term is generated by the mass flux in the pressure adjustment process (Neale et al., 2012, accessed July 02, 2021; Lauritzen et al., 2022), and the L term is generated separately from the mass flux by a macrophysics package responsible for surface flux absorption.

The pressure adjustment process is energy conserving, a constraint enforced by a dynamical core (dycore) energy fixer (Lauritzen and Williamson, 2019). Therefore, instead of the total flux in (2), the atmosphere receives only amount $(L_v + L_l)\Delta m$.

Separately, a variable called 'latent heat' (LH), defined as $LH = L_v \Delta m$, is used to compute the ocean temperature tendency via

$$\Delta T_o = T_o^{new} - T_o = -\frac{LH}{c_l m_o^l} = -\frac{L_v \Delta m}{c_l m_o^l}, \tag{3}$$

where superscript 'new' denotes the post-evaporation temperature.

This temperature tendency is applied to the remaining ocean mass $m_o^l$, leading to an updated ocean energy after evaporation:

$$E_{ocean}^{new} = m_o^l(c_l T_o^{new} + L_l) = c_l m_o^l \left(T_o - \frac{L_v \Delta m}{c_l m_o^l}\right) + L_l m_o^l = c_l T_o m_o^l + L_l m_o^l - L_v \Delta m.$$

Thus, after evaporation (incorporating the actions of pressure adjustment, fixer, and new temperature tendency), the atmosphere gains energy $(L_v + L_l)\Delta m$, while the ocean loses $c_l T_o \Delta m + (L_v + L_l)\Delta m$. The total energy loss from the ocean exceeds the gain by the atmosphere by $c_l T_o \Delta m$, which is unaccounted for in the energy budget. This missing energy is compensated by the fixer IEFLX (Golaz et al., 2019), which injects $c_l T_o \Delta m$ into the atmosphere to restore global energy balance. In the implementation of the operational models, this artificial balance is applied by distributing the missing energy equally among all grid cells used to represent the atmosphere, i.e., IEFLX is a global fixer.





### 2.2.3 Proposed new model

As shown in the previous section, the current E3SM design relies on implicit energy flux assumptions, inconsistent thermodynamics between components, and various energy fixers—all of which complicate the model and render the thermodynamics at the ocean-atmosphere interface physically inconsistent. Here we propose an alternative and possibly improved framework to address evaporation at this interface.

     The liquid water thermodynamics is still given by (1), but the vapor energy (enthalpy) associated with the evaporative flux
is now:

$$F_{atm} = c_p^v T \Delta m + (L_v + L_l)\Delta m. \tag{4}$$

Here, the phase change occurs in the ocean, and the energy required for vaporization is withdrawn from the ocean itself. This can be expressed as a conservation of energy equation, where the left hand side is the energy of the ocean before the phase change and the the right hand side is the energy after:

$$c_l T_o (m_o^l + \Delta m) + L_l(m_o^l + \Delta m) = c_l m_o^l T_o^{new} + c_p^v \Delta m T_o + L_l m_o^l + (L_l + L_v)\Delta m. \tag{5}$$

The details on whether to assign the vapor parcel temperature $T_o$ or $T_o^{new}$ are discussed later in Sec. 3.4. Rearranging Eq. (5) yields a new temperature tendency for the ocean:

$$\Delta T_o = T_o^{new} - T_o = \frac{(c_l T_o \Delta m + L_l \Delta m) - (c_p^v T_o \Delta m + (L_v + L_l)\Delta m)}{c_l m_o^l} = \frac{(c_l - c_p^v)T_o \Delta m - L_v \Delta m}{c_l m_o^l}. \tag{6}$$

Notably, the temperature tendency in this equation is different from that in (3) representing the current E3SM design.

This approach is both energy-conserving, as the atmosphere receives the full energy flux $c_p^v T_o \Delta m + (L_v + L_l)\Delta m$ from the ocean and physically grounded, since there are no fixers involved.

     In both the current (equation (3)) and the proposed (equation (6)) models, the ocean temperature tendency is proportional to the *latent heat of vaporization*, defined as the difference in enthalpy between vapor and liquid forms of same mass $\Delta m$. However, in the current model, these enthalpies are defined using dry-air heat capacity ($c_p^d$), while the proposed model uses
species-appropriate heat capacities. These conceptual differences are summarized in Table 1. We observe from comparing the current and the proposed design that during evaporation, in the current model, the atmosphere receives and the ocean loses more energy than in the proposed model. Later in Section 4 we will show that condensation triggers an opposite behavior in the energy deficit/ excess in the current model. However, the magnitude of errors (measured as differences between the current model and the model with unapproximated thermodynamics) during condensation are smaller than those during evaporation.

### 2.2.4 Bulk methods do not capture energy transfers from water phase changes

Surface stress fluxes, often represented using Monin–Obukhov Similarity Theory (MOST), are typically modeled using bulk formulations of the form:

$$F_X = \rho\, C_X (X_z - X_{surf}),$$





| evaporation model | energy flux received by atmosphere | ocean T tendency | definition of latent heat of vaporization, LH | enthalpy of vapor in atmosphere | enthalpy of liquid in atmosphere |
|---|---|---|---|---|---|
| current | $c_l T \Delta m + (L_v + L_l)\Delta m$ | $\Delta T = -\frac{LH}{c_l m^l}$ | $LH = h^v_{approx.} - h^l_{approx.}$ | $h^v_{approx.} = (c^d_p T + L_v + L_l)\Delta m$ | $h^l_{approx.} = (c^d_p T + L_l)\Delta m$ |
| proposed | $c^v_p T \Delta m + (L_v + L_l)\Delta m$ | $\Delta T = -\frac{LH}{c_l m^l}$ | $LH = h^v - h^l$ | $h^v = (c^v_p T + L_v + L_l)\Delta m$ | $h^l = (c_l T + L_l)\Delta m$ |

**Table 1.** Conceptual summary of the current and proposed implementations of evaporation

where $F_X$ is a flux of quantity $X$ (temperature, a velocity component, or vapor), $\rho$ is the air density, $C_X$ is a combination of transfer coefficients and bulk expressions, and $X_z$ and $X_{surf}$ represent the value of the variable $X$ at some reference height, $z$, and at the surface, respectively (Fairall et al., 1996; Taylor, 2015).

The key point of our work is that while these bulk schemes compute a mass flux of vapor and a heat flux, they do not explicitly model the heat transfers during evaporation. This differs from the treatment in atmospheric physics parametrizations (e.g., evaporated rain), where energy (or enthalpy) conservation due to phase changes is modeled explicitly (Lauritzen et al., 2022; Guba et al., 2024), as well as from the formulation of evaporation we present in Sections 2.2.2 and 2.2.3.

## 3 Thermodynamics of phase change and simplified models of ocean-atmosphere water exchanges

In this section, we examine the exchange of water between the ocean and atmosphere in more detail. The full process of converting atmospheric water vapor into oceanic liquid via precipitation is referred to as condensation. Condensation encapsulates a two-part process. The first stage occurs entirely within the atmosphere, where water vapor condenses into droplets—represented by the top/grey portion of the left panel in Fig. 1. The second stage, shown as grey boxes in the right panel, corresponds to the sedimentation or precipitation of these droplets into the ocean.

Similarly, evaporation is also conceptualized as a two-stage process, illustrated by the blue regions in Fig. 1. When water evaporates from the ocean, it first becomes water vapor within the ocean before subsequently ascending into the atmosphere.

This two-part decomposition of each phase-change process—both from atmosphere to ocean (via precipitation) and from ocean to atmosphere (via evaporation)—is not merely schematic. It is essential for correctly incorporating unapproximated thermodynamics. By distinguishing between the stages, we ensure that the appropriate specific heat capacity is used for the relevant water form (liquid or vapor) at each step.

For example, during Stage 1 of each process, the latent heat exchange occurs within the originating component: in the atmosphere during condensation, and in the ocean during evaporation. This perspective aligns with the discussion of enthalpy and energy partitioning presented earlier in Section 2.2.3.





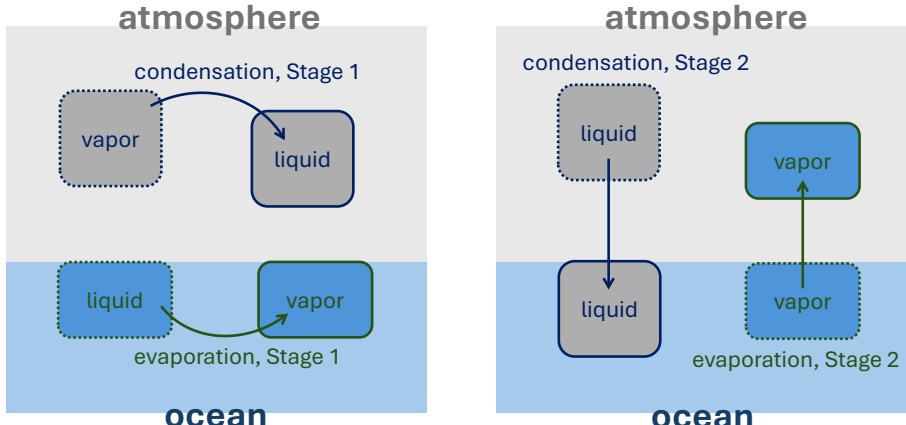

**Figure 1.** Schematics for the two stages in condensation and evaporation processes: Stage 1 is a phase change within the component, Stage 2 is a transfer of a water species flux to the other component.

This careful dissection of each leg of the water exchange process directly enables the derivation of unapproximated thermodynamics. It also provides a clear basis for identifying deficiencies in the current E3SM implementation of moist thermodynamics. Within this framework, the time rate of tendencies of atmospheric and oceanic temperature and water mass are formulated as a system of ordinary differential equations in time. This allows us to systematically examine and compare the evolution of the ocean–atmosphere system under both the proposed formulation and the existing E3SM design. We begin with the proposed formulation.

### 3.1 Unapproximated thermodynamics

In this section we start with deriving equations for tendencies of water mass and temperature in the ocean and the atmosphere. It leads to a system of four coupled algebraic equations. Both components, the atmosphere and the ocean, are modeled as simple dimensionless boxes. The mass variables are defined as follows: $m_a^v$, $m_a^l$, $m_a^d$ denote atmospheric water vapor, liquid, and dry air, respectively, while $m_o^l$ and $m_o^v$ represent oceanic liquid and vapor mass. The temperatures of the atmosphere and the ocean are given by $T_a$ and $T_o$, respectively. Each variable represents a mean value over a single grid cell or box. The guiding principle for the equations below is **conservation of mass and energy** after each process (stage). These processes include phase changes (vapor→liquid and liquid→vapor), sedimentation of the atmospheric water liquid into the ocean, and transfer of the evaporated ocean vapor into the atmosphere. Each process updates the initial quantities (unmarked) to new values (denoted with superscript 'new'). For example, a mass change in atmospheric water vapor content is written as $\Delta m_v^a := (m_a^v)^{new} - m_a^v$. For simplicity, we will use such $\Delta$ notation as much as possible.

These algebraic equations capture the instantaneous changes in mass and temperature associated with prescribed evaporation and condensation amounts, denoted by $\Delta V$ and $\Delta K$, respectively. As these evaporation and condensation rates are assumed





to be known, it is useful to convert the algebraic system into time-dependent equations. This leads to a system of ordinary differential equations (ODEs) governing the evolution of atmospheric and oceanic mass and temperature. The derivation of these ODEs is presented in section 3.4.

The total energy is $E_{atm} + E_{ocn}$, where the energy of the atmosphere and ocean is respectively,

$$E_{atm} = (c_p^d m_a^d + c_p^v m_a^v)T_a + (L_v + L_l)m_a^v, \quad E_{ocn} = c_l T_o m_o^l + L_l m_o. \tag{7}$$

The first terms in $E_{atm}$ and $E_{ocn}$, involving heat capacities multiplied with temperature are commonly referred to as **enthalpies** in the literature. In older formulations, enthalpy is frequently defined without the $L$ terms, treating the corresponding energy released or absorbed during phase transitions of water as external inputs. Such an approach complicates energy conservation in
a model due to an increased requirement of book-keeping. Therefore, we adhere to definitions of enthalpy that include $L$ terms, like in Thuburn (2017) and Eldred et al. (2022). Sometimes, in this work, we operate with energy defined by the L-terms. We may refer to this energy as *latent heat internal energy*. Let us now consider the required tendencies in the condensation process, before addressing the evaporation.

### 3.1.1   Condensation

As schematized earlier in the grey portion of Fig. 1a during the first stage in the condensation process, a phase change occurs such that a mass $\Delta K > 0$ of water vapor undergoes phase change to become liquid while remaining suspended in the atmosphere. Therefore, the mass balance is

$$\Delta m_a^v = -\Delta K, \quad \Delta m_a^l = \Delta K, \quad \Delta m_o^l = 0, \quad \Delta m_o^v = 0.$$

Before the phase change, the condensing vapor has specific heat capacity $c_p^v$ and latent heat internal energy $L_v + L_l$. After the phase change, the resulting liquid has heat capacity $c_l$ and latent energy $L_v$. Assuming the atmospheric temperature changes from $T_a$ to $T_a^{new}$, the energy conservation is formulated as:

$$\underbrace{(c_p^v(m_a^v + \Delta K) + c_p^d m_a^d)T_a + (L_v + L_l)(m_a^v + \Delta K)}_{\text{energy of atm. before phase change}} = \underbrace{(c_p^v m_a^v + c_l \Delta K + c_p^d m_a^d)T_a^{new} + (L_v + L_l)m_a^v + L_l \Delta K}_{\text{energy of atm. after phase change}}.$$

Since no change occurs in the ocean during this stage (denoting grey phase during condensation in Fig. 1, the ocean temperature satisfies $\Delta T_o = 0$.

In the second stage of the condensation process, the newly formed liquid is removed from the atmosphere and deposited into the ocean. The atmosphere temperature $T_a$ does not change during this stage. The necessary changes to $T_a$ accompanying condensation related phase change were already included in first stage. The mass conservation in this state implies:

$$\Delta m_a^v = 0, \quad \Delta m_a^l = -\Delta K, \quad \Delta m_o^l = \Delta K, \quad \Delta m_o^v = 0.$$

The conservation of energy in the ocean leads to:

$$\underbrace{c_l T_o m_o^l + L_l m_o^l}_{\text{energy of ocn. before precip.}} + \underbrace{c_l T_a^{new} \Delta K + L_l \Delta K}_{\text{energy of precip.}} = \underbrace{c_l T_o^{new}(m_o + \Delta K) + L_l(m_o^l + \Delta K)}_{\text{energy of ocn. after precip.}}.$$





Combining both stages and eliminating $m_a^l$ and $m_o^v$, we obtain

$$\Delta m_a^v = -\Delta K \tag{8}$$

$$\Delta m_o^l = \Delta K \tag{9}$$

$$(c_p^v(m_a^v + \Delta K) + c_p^d m_a^d)T_a + (L_v + L_l)(m_a^v + \Delta K) = \tag{10}$$
$$(c_p^v m_a^v + c_l \Delta K + c_p^d m_a^d)T_a^{new} + (L_v + L_l)m_a^v + L_l \Delta K$$

$$c_l m_o^l T_o + L_l m_o^l + c_l T_a^{new}\Delta K + L_l \Delta K = c_l T_o^{new}(m_o^l + \Delta K) + L_l(m_o^l + \Delta K) \tag{11}$$

Converting these algebraic equations into ODEs involves additional assumptions, which we discuss later in Sec. 3.4.1.

### 3.1.2 Evaporation

The required mass and temperature tendencies of the ocean and atmosphere during evaporation follow a derivation closely analogous to that presented above for condensation. This is detailed in the current subsection. The first stage is the phase change of mass $\Delta V > 0$ of oceanic liquid water into vapor, while it remains in the ocean. This representation is essential to correctly model ocean cooling due to latent heat loss. Mass conservation is given by

$$\Delta m_a^v = 0, \quad \Delta m_a^l = 0, \quad \Delta m_o^l = -\Delta V, \quad \Delta m_o^v = \Delta V.$$

Initially, the evaporating liquid has specific heat capacity $c_l$ and latent heat internal energy defined by $L_l$. After the phase change following evaporation these change to $c_p^v$ and $L_v + L_l$. The energy conservation for this phase change in the ocean is given by

$$\underbrace{c_l T_o(m_o^l + \Delta V) + L_l(m_o^l + \Delta V)}_{\text{energy of ocn. before phase change}} = \underbrace{(c_l m_o^l + c_p^v \Delta V)T_o^{new} + L_l m_o^l + (L_v + L_l)\Delta V}_{\text{energy of ocn. after phase change}} . \tag{12}$$

In the second stage, the vapor leaves the ocean and becomes a part of the atmosphere, following a mass conservation given by

$$\Delta m_a^v = \Delta V, \quad \Delta m_a^l = 0, \quad \Delta m_o^l = 0, \quad \Delta m_o^v = -\Delta V.$$

The energy conservation in the atmosphere is:

$$\underbrace{(c_p^d m_a^d + c_p^v m_a^v)T_a + (L_v + L_l)m_a^v}_{\text{energy of atm. before evap.}} + \underbrace{c_p^v T_o^{new}\Delta V + (L_v + L_l)\Delta V}_{\text{energy of evap. flux}} = \underbrace{(c_p^d m_a^d + c_p^v m_a^v + c_p^v \Delta V)T_a^{new} + (L_v + L_l)(m_a^v + \Delta V)}_{\text{energy of atm. after evap.}} .$$

Combining both stages, we obtain

$$\Delta m_a^v = \Delta V, \tag{13}$$

$$\Delta m_o^l = -\Delta V, \tag{14}$$

$$c_l T_o(m_o^l + \Delta V) + L_l(m_o^l + \Delta V) = (c_l m_o^l + c_p^v \Delta V)T_o^{new} + L_l m_o^l + (L_v + L_l)\Delta V, \tag{15}$$

$$(c_p^d m_a^d + c_p^v m_a^v)T_a + c_p^v T_o^{new}\Delta V + (L_v + L_l)(m_a^v + \Delta V) = (c_p^d m_a^d + c_p^v m_a^v + c_p^v \Delta V)T_a^{new} + (L_v + L_l)(m_a^v + \Delta V). \tag{16}$$





## 3.2 Current E3SM implementation

While the E3SM surface exchange thermodynamics were likely not derived in the manner presented here, we reinterpret them using the same framework applied to the unapproximated formulation in the previous section. Here, both evaporation and condensation processes require additional steps that represent pressure adjustment, its energy fixer, and the IEFLX energy fixer previously discussed in section 2.2.2.

In this formulation, the total atmospheric energy differs from that in the unapproximated case, Eq. (7), but the ocean energy
remains the same:

$$E_{atm} = c_p^d(m_a^d + m_a^v)T_a + (L_v + L_l)m_a^v, \quad E_{ocn} = c_l T_o m_o^l + L_l m_o. \tag{17}$$

The difference in atmospheric energy arises from the use of the dry-air specific heat capacity, $c_p^d$, being applied to the atmospheric vapor mass $m_a^v$, instead of the vapor-specific heat capacity $c_p^v$, used previously in equation (7). Such discrepancies in specific heat capacities constitute one of the primary sources of divergence between the unapproximated formulation and
the current E3SM design. As we will demonstrate in section 4, these are not merely minor quantitative errors—they result in significant qualitative differences in the (simplified) system's behavior.

### 3.2.1 Condensation

Since in E3SM energy flux of precipitation is not modeled explicitly, it is represented instead by a few processes, as described below. To clearly explain the mechanism of precipitation, in this section we need to operate with one more time index. Besides
$T_a$ and $T_a^{new}$, we introduce intermediate $T_a^{new'}$.

The first stage in the condensation process remains structurally the same as in the unapproximated case of the previous section, but now with heat capacities of the dry air for water forms in the atmosphere. The mass conservation constitutes

$$\Delta m_a^v = -\Delta K, \quad \Delta m_a^l = \Delta K, \quad \Delta m_o^l = 0, \quad \Delta m_o^v = 0,$$

and the energy conservation during the phase change within the atmosphere is

$$\underbrace{c_p^d(m_a^v + m_a^d + \Delta K)T_a + (L_v + L_l)(m_a^v + \Delta K)}_{\text{energy of atm. before phase change}} = \underbrace{c_p^d(m_a^v + m_a^d + \Delta K)T_a^{new'} + (L_v + L_l)m_a^v + L_l\Delta K}_{\text{energy of atm. after phase change}}. \tag{18}$$

In the unapproximated case, the sedimentation, included in the second stage of condensation process, did not alter the atmospheric temperature $T_a$, because the energy flux associated with the precipitating mass $\Delta K$ was matched between the
ocean and the atmosphere. In E3SM, however, the vapor-to-liquid transition followed by sedimentation is not modeled with consistent energy (or enthalpy) fluxes.

Specifically, in EAM, the energy associated with mass $\Delta K$ during sedimentation is given by $L_l\Delta K$ and $E_1 = c_p^d T_a^{new'}\Delta K$. As described by Neale et al. (2012, accessed July 02, 2021) and Lauritzen et al. (2022), during the pressure adjustment process, energy $E_1$ is removed from the atmosphere, but then restored by the dynamical core energy fixer (Lauritzen et al., 2022),



ensuring energy conservation in the pressure adjustment process. As a result, the net outgoing energy flux from the atmosphere into the ocean is only $L_l \Delta K$.

This is further corrected by IEFLX energy fixer, which removes energy $E_2 = c_l T_a^{new} \Delta K$ (or $E_2 = c_l T_a \Delta K$ as discussed later in Sec. 3.4.1) from the atmosphere via temperature globally. This action restores the correct outgoing energy flux of precipitation to value $E_2 + L_l \Delta K$, which is taken up by the ocean. In our simple box model, after incorporating the net energy transfers, the conservation of energy in the atmosphere and the ocean during the precipitation/sedimentation, i.e., the second stage of condensation process is given by

$$\underbrace{c_p^d(m_a^v + m_a^d + \Delta K)T_a^{new'} + (L_v + L_l)m_a^v + L_l \Delta K}_{\text{energy of atm. before precip.}} = \underbrace{c_p^d(m_a^v + m_a^d)T_a^{new} + (L_v + L_l)m_a^v}_{\text{energy of atm. after precip.}} + \underbrace{E_2 + L_l \Delta K}_{\text{energy of precip.}} \tag{19}$$

$$\underbrace{c_l T_o m_o^l + L_l m_o^l}_{\text{energy of ocn. before precip.}} + \underbrace{E_2 + L_l \Delta K}_{\text{energy of precip.}} = \underbrace{c_l T_o^{new}(m_o^l + \Delta K) + L_l(m_o^l + \Delta K)}_{\text{energy of ocn. after precip.}} \tag{20}$$

We re-derive (18) and (19) as one equation:

$$\underbrace{c_p^d(m_a^d + m_a^v + \Delta K)T_a + (L_l + L_v)(m_a^v + \Delta K)}_{\text{energy of atm. before precip.}} = \underbrace{c_p^d(m_a^d + m_a^v)T_a^{new} + (L_v + L_l)m_a^v}_{\text{energy of atm. after precip.}} + \underbrace{c_l T_a^{new}\Delta K + L_l \Delta K}_{\text{energy of precip.}}$$

Similar to the unapproximated case, combining both stages, we obtain

$$\Delta m_a^v = -\Delta K \tag{21}$$

$$\Delta m_o^l = \Delta K \tag{22}$$

$$c_p^d(m_a^d + m_a^v + \Delta K)T_a + (L_l + L_v)(m_a^v + \Delta K) = c_p^d(m_a^d + m_a^v)T_a^{new} + (L_v + L_l)m_a^v + c_l T_a^{new}\Delta K + L_l \Delta K \tag{23}$$

$$c_l(T_o m_o^l + T_a^{new}\Delta K) + L_l(m_o^l + \Delta K) = c_l T_o^{new}(m_o^l + \Delta K) + L_l(m_o^l + \Delta K) \tag{24}$$

### 3.2.2 Evaporation

In the current E3SM design, the first stage of the evaporation process incorporating the phase change from liquid to evaporated state within the ocean is implemented via a temperature tendency directly proportional to latent energy of vaporization:

$$\Delta T_o = -\frac{L_v \Delta V}{c_l m_o^l} \ .$$

This implies the following energy balance equation in the ocean during the phase change process (first stage):

$$\underbrace{c_l T_o m_l + c_l T_o \Delta V + L_l(m_l + \Delta V)}_{\text{energy of ocn. before phase change}} = \underbrace{(c_l m_l + c_l \Delta V)T_o^{new} + L_l m_l + (L_v + L_l)\Delta V}_{\text{energy of ocn. after phase change}} . \tag{25}$$

The difference from the unapproximated case (compare Eq. (25) with Eq. (12)) lies in the use of $c_l$ (liquid) heat capacity instead of more appropriate $c_p^v$ (vapor) for the evaporated mass.

In the second stage, vapor leaves the ocean and becomes a part of the atmosphere. As with condensation, the pressure adjustment and the dynamical core energy fixer ensure no net atmospheric energy change from this part of the process except





for the L term $(L_v + L_l)\Delta V$. The full energy of the incoming vapor mass into the atmosphere is thus corrected with IEFLX term $E_3 = c_l T_o^{new} \Delta V$. This leads to the following equation for the conservation of energy in the atmosphere:

$$\underbrace{c_p^d(m_a^d + m_a^v)T_a + (L_v + L_l)m_a^v}_{\text{energy of atm. before evap. flux}} + \underbrace{E_3 + (L_v + L_l)\Delta V}_{\text{energy of evap. flux}} = \underbrace{c_p^d(m_a^d + m_a^v + \Delta V)T_a^{new} + (L_v + L_l)(m_a^v + \Delta V)}_{\text{energy of atm. after evap. flux}} \ .$$

Unlike condensation, no additional tendency is applied to $T_o$ in the second stage, since the ocean has already lost energy flux

$E_3 + (L_v + L_l)\Delta V = c_l T_o^{new} \Delta V + (L_v + L_l)\Delta V$ represented in (25).

As before, combining the two states, the equations comprising evaporation in E3SM are

$$\Delta m_a^v = \Delta V, \tag{26}$$

$$\Delta m_o^l = -\Delta V, \tag{27}$$

$$c_l T_o m_l + c_l T_o \Delta V + L_l(m_l + \Delta V) = (c_l m_l + c_l \Delta V)T_o^{new} + L_l(m_l + \Delta V) + L_v \Delta V, \tag{28}$$

$c_p^d(m_a^d + m_a^v)T_a + (L_v + L_l)m_a^v + c_l T_o^{new}\Delta V + (L_v + L_l)\Delta V =$

$$c_p^d(m_a^d + m_a^v + \Delta V)T_a^{new} + (L_v + L_l)(m_a^v + \Delta V). \tag{29}$$

The framework presented in this section follows the E3SM formulation, with one box representing the atmosphere and one for the ocean. In this simplified setting—where the distinction between local and global behavior is blurred—the various energy fixers can be interpreted as acting "locally" to each grid cell (just one in our simplified box model). Local energy fixers

are known to be detrimental to model fidelity and predictive accuracy (Harrop et al., 2022). In fact, it may be preferable to relax strict energy conservation altogether if globally consistent fixers cannot be applied. Motivated by this, we introduce an alternative model in the next section: an E3SM-like formulation that forgoes net energy conservation.

### 3.3 Model with E3SM-like behavior (no local fixers)

The systems described by Eqs.(8)–(11), (13)–(16) (for the unapproximated case) and Eqs.(21)–(24), (26)–(29) (for the current

E3SM design) represent significant simplifications relative to the full complexity of E3SM. In the actual E3SM implementation, IEFLX terms, partially responsible for the nonphysical behavior observed in the simplified version of current model discussed later in Section 4, are applied as global fixers. These are implemented as globally integrated energy corrections that result in the same temperature tendency at each horizontal grid cell at each vertical model level. Importantly, because the evaporation and precipitation fluxes are approximately balanced (globally) in E3SM, the magnitude of these temperature corrections (due

to both IEFLX and the dynamical core energy fixer) is relatively small at each grid point.

Since the fixers in the actual E3SM simulations lead to small temperature tendencies, we modify the equations (23)–(24), (28)–(29) for current E3SM implementation and remove the effect of the IEFLX and the dynamical core fixer. This model maintains the basic thermodynamic structure of E3SM but relaxes net energy conservation. Because the pressure adjustment in E3SM does not modify the atmospheric temperature $T_a$, there is no temperature tendency from that process. Accordingly, we modify the current model's equations as follows. For condensation, we rewrite Eq. (23) as

$$c_p^d(m_a^d + m_a^v + K)T_a + (L_l + L_v)(m_a^v + K) = c_p^d(m_a^d + m_a^v)T_a^{new} + (L_v + L_l)m_a^v + c_p^d T_a^{new} K + L_l K \ .$$





For evaporation, the current design of E3SM implies that there is no temperature tendency for $T_a$ due to the incoming evaporative flux. Thus, there is no atmospheric energy equation analogous to Eq. (29) and no corresponding correction to $T_a$ arising from $\Delta V$.

### 3.4 From tendency (algebraic) equations to time derivatives (ODEs)

#### 3.4.1 Considerations

Now we reformulate the systems of algebraic equations representing the tendencies of oceanic and atmospheric mass and temperature from above as systems of ordinary differential equations representing the time rate of these tendencies. We begin with the unapproximated condensation model defined by (8)–(11) and outline the assumptions used in this reformulation in detail.

Consider Eq. (8). Introducing a finite time step $\Delta t$ over which the change $\Delta m_a^v$ occurs, we arrive at:

$$\Delta m_a^v = -\Delta K \quad \Rightarrow \quad \frac{\Delta m_a^v}{\Delta t} = -\frac{\Delta K}{\Delta t} \quad \Rightarrow \quad \frac{dm_a^v}{dt} = -\frac{dK}{dt},$$

where $\dfrac{dK}{dt}$ is the condensation rate, to be defined later. Now consider the energy balance from Eq. (10), repeated below:

$$(c_p^v(m_a^v + \Delta K) + c_p^d m_a^d)T_a + (L_v + L_l)(m_a^v + \Delta K) = (c_p^v m_a^v + c_l \Delta K + c_p^d m_a^d)T_a^{new} + (L_v + L_l)m_a^v + L_l \Delta K.$$

To the leading order in $\Delta K$ we can express the atmospheric temperature tendency as:

$$T_a^{new} - T_a = \Delta T_a = \frac{c_p^v T_a - c_l T_a + L_v}{c_p^d m_a^d + c_p^v m_a^v + c_l \Delta K}\Delta K = \frac{c_p^v T_a - c_l T_a + L_v}{c_p^d m_a^d + c_p^v m_a^v}\Delta K + O((\Delta K)^2). \tag{30}$$

Dividing this by $\Delta t$, taking the limit, $\Delta t \to 0$, and considering only the first order terms in the condensation rate, $dK/dt = \lim_{\Delta t \to 0}(\Delta K/\Delta t)$, we obtain the corresponding ODE for atmospheric temperature,

$$\frac{dT_a}{dt} = \frac{c_p^v T_a - c_l T_a + L_v}{c_p^d m_a^d + c_p^v m_a^v}\frac{dK}{dt}. \tag{31}$$

This approach is applied systematically to all the variables in both condensation and evaporation processes to derive the full set of time-dependent governing equations for our simplified box model. The resulting systems of ODEs are presented in the following three subsections, corresponding to each of the three models: the unapproximated model (labeled as System I for Ideal), the current E3SM implementation (labeled as System A1 for First Approximation), and the E3SM-like model without energy fixers (labeled as System A2 for Second Approximation).





### 3.4.2 The final ODE system for the ideal case: System I

Analogously to the steps above, we convert the entire algebraic system (8)–(11) into the system of ODEs

$$\frac{dm_a^v}{dt} = -\frac{dK}{dt}, \tag{32}$$

$$\frac{dm_o^l}{dt} = \frac{dK}{dt}, \tag{33}$$

$$\frac{dT_a}{dt} = \frac{c_p^v T_a - c_l T_a + L_v}{c_p^d m_a^d + c_p^v m_a^v} \frac{dK}{dt}, \tag{34}$$

$$\frac{dT_o}{dt} = \frac{T_a - T_o}{m_l^o} \frac{dK}{dt} \tag{35}$$

From the evaporation equations (13)–(16) we similarly obtain:

$$\frac{dm_a^v}{dt} = \frac{dV}{dt}, \tag{36}$$

$$\frac{dm_o^l}{dt} = -\frac{dV}{dt}, \tag{37}$$

$$\frac{dT_a}{dt} = \frac{c_p^v(T_o - T_a)}{c_p^d m_a^d + c_p^v m_a^v} \frac{dV}{dt}, \tag{38}$$

$$\frac{dT_o}{dt} = \frac{c_l T_o - c_p^v T_o - L_v}{c_l m_o^l} \frac{dV}{dt} \tag{39}$$

Combining both condensation and evaporation, the full system becomes

$$\frac{dm_a^v}{dt} = -\frac{dK}{dt} + \frac{dV}{dt}, \tag{40}$$

$$\frac{dm_o^l}{dt} = \frac{dK}{dt} - \frac{dV}{dt}, \tag{41}$$

$$\frac{dT_a}{dt} = \frac{c_p^v T_a - c_l T_a + L_v}{c_p^d m_a^d + c_p^v m_a^v} \frac{dK}{dt} + \frac{c_p^v(T_o - T_a)}{c_p^d m_a^d + c_p^v m_a^v} \frac{dV}{dt}, \tag{42}$$

$$\frac{dT_o}{dt} = \frac{T_a - T_o}{m_l^o} \frac{dK}{dt} + \frac{c_l T_o - c_p^v T_o - L_v}{c_l m_o^l} \frac{dV}{dt} . \tag{43}$$

### 3.4.3 The final ODE system for the current case: System A1

Applying the same procedure to the algebraic systems (21)–(24) and (26)–(29), we obtain

$$\frac{dm_a^v}{dt} = -\frac{dK}{dt} + \frac{dV}{dt}, \tag{44}$$

$$\frac{dm_o^l}{dt} = \frac{dK}{dt} - \frac{dV}{dt}, \tag{45}$$

$$\frac{dT_a}{dt} = \frac{c_p^d T_a - c_l T_a + L_v}{c_p^d(m_a^d + m_a^v)} \frac{dK}{dt} + \frac{c_l T_o - c_p^d T_a}{c_p^d(m_a^d + m_a^v)} \frac{dV}{dt}, \tag{46}$$

$$\frac{dT_o}{dt} = \frac{T_a - T_o}{m_o^l} \frac{dK}{dt} - \frac{L_v}{c_l m_o^l} \frac{dV}{dt} . \tag{47}$$

One can verify that systems (40)–(43) and (44)–(47) are energy-conserving in the sense that $\frac{d(E_{atm} + E_{ocn})}{dt} = 0$.




### 3.4.4 The final ODE system for the E3SM-like case: System A2

Here, the energy fixers are omitted, and we obtain:

$$\frac{dm_a^v}{dt} = -\frac{dK}{dt} + \frac{dV}{dt}, \tag{48}$$

$$\frac{dm_o^l}{dt} = \frac{dK}{dt} - \frac{dV}{dt}, \tag{49}$$

$$\frac{dT_a}{dt} = \frac{L_v}{c_p^d(m_a^d + m_a^v)}\frac{dK}{dt}, \tag{50}$$

$$\frac{dT_o}{dt} = \frac{T_a - T_o}{m_o^l}\frac{dK}{dt} - \frac{L_v}{c_l m_o^l}\frac{dV}{dt}. \tag{51}$$

Note that this system does not conserve energy (17) due to the omission of the IEFLX and dynamical core fixers.

### 3.4.5 Atmospheric temperature tendency due to evaporation

We draw attention to the $\frac{dV}{dt}$ term in $\frac{dT_a}{dt}$ equations. In the three systems, it appears as

– System I:

$$\frac{c_p^v(T_o - T_a)}{c_p^d m_a^d + c_p^v m_a^v}\frac{dV}{dt},$$

– System A1:

$$\frac{c_l T_o - c_p^d T_a}{c_p^d(m_a^d + m_a^v)}\frac{dV}{dt},$$

– System A2: Zero (no contribution to $\frac{dT_a}{dt}$ from evaporation).

Therefore, in System I, the temperature tendency depends on the temperature difference at the ocean-atmosphere interface, which in physically reasonable. In contrast, since $c_l \approx 4c_p^d$. System A1 yields positive tendency in atmospheric temperature for realistic values of $T_a$ and $T_o$, regardless of the sign of difference $T_o - T_a$, which we regard as physically unrealistic. In System A2, the absence of any $\frac{dT_a}{dt}$ tendency due to evaporation is also implausible. Below in Sec. 4 we show that the physically unrealistic $\frac{dT_a}{dt}$ term in System A1 contributes to the system's unstable behavior.

### 3.4.6 Three systems and their relation to E3SM

With the simplified models for thermodynamic exchange at the ocean–atmosphere interface in place—System I (ideal with unapproximated theromodynamics), System A1 (E3SM-like with fixers), and System A2 (E3SM-like without fixers)—we now clarify their correspondence to E3SM and the assumptions involved.

If we consider the whole Earth system to be presented by two boxes, one for the ocean and one for the atmosphere, just like we outlined in Sec. 3.1, then System A1 is an appropriate representation of E3SM, while System A2 is not, since it does not





conserve energy. However, E3SM consists of many degrees of freedom, effectively many vertical columns, and in that context, each column's thermodynamic treatment aligns more closely with System A2, before energy fixers are applied.

These energy fixers in E3SM simulations are relatively small in magnitude. Their values can be approximated via $c_l T_{surf}(P-Q)$ for IEFLX (Golaz et al., 2019) and $c_p^d T_{surf}(P-Q)$ for the dycore fixer (Lauritzen et al., 2022), where $P$ and $Q$ are globaly integrated precipitation and evaporation rates, respectively. In time-averaged multi-seasonal runs, $P \approx Q$ within about $1 \times 10^{-8}$ kg m$^{-2}$ sec$^{-1}$. Thus, an ensemble of system A2 models, each representing a vertical column, is a valid proxy for E3SM as long as global precipitation and evaporation approximately balance. This ensemble would require a small global energy correction

of the order $3 \times 10^{-3}$ to $1.2 \times 10^{-2}$ W m$^{-2}$.

     By contrast, System A1 does not accurately describe E3SM's behavior when modeling multiple columns, as it effectively implements a local energy fixer. Such local fixers have been shown to degrade performance (Harrop et al., 2022).

     System I, unlike A1 and A2, does not require external assumptions about energy fixers or balance of precipitation and evaporation. It can be used in either the global box model setting or in multi-column settings, and it always conserves energy

since that is built into it from the first principles. Furthermore, unlike current design of E3SM, it does not require an implicit requirement of $P \approx Q$.

### 3.4.7   Evaporation and condensation rates

In all systems, we define the evaporation and condensation rates as

$$\frac{dV}{dt}(m_a^v, T_a) \;=\; C_h |u| \frac{1}{z_a} \max\{(m_a^d + m_a^v)q_{\text{sat}}(T_a) - m_a^v, 0\}, \tag{52}$$

$$\frac{dK}{dt}(m_a^v, T_a) \;=\; \lambda \max\{m_a^v - (m_a^d + m_a^v)q_{\text{sat}}(T_a), 0\}, \tag{53}$$

$$q_{\text{sat}}(T) \;=\; \frac{c_1 \cdot c_2}{p_0} \exp\left(-\frac{L_v}{R_v}\left(\frac{1}{T} - \frac{1}{T_0}\right)\right), \tag{54}$$

where the constants are: $C_h = 0.0011$, $|u| = 50.0$ m sec$^{-1}$, $z_a = 50.0$ m, $\lambda = 1/100$ sec$^{-1}$, $c_1 = 0.622$, $c_2 = 610.78$ Pa, $p_0 = 10^5$ Pa, $R_v = 461$ J kg$^{-1}$ K$^{-1}$, and $T_0 = 273.16$ K. The mass of the dry air in simulations below is fixed to $m_a^d = 1000$ kg and ocean's mass is initialized to 5000 kg in all cases presented in the next section.

In the next section, we use MATLAB's numerical ODE integration tools to simulate the evolution of atmospheric and oceanic mass and temperature, in order to evaluate the performance of the three models: System I, System A1, and System A2.

     Since $q_{\text{sat}}$ is well defined away from $T = 0$, functions (52) and (53) are continuous and Lipschitz-continuous away from $T = 0$ K, ensuring existence and uniqueness of solutions under physically relevant conditions (away from $T = 0$ K).

## 4   Numerical analysis of Systems I, A1, and A2

Since the evaporation and condensation rates depend oppositely on the sign of the expression $(m_a^v + m_a^d)q_{\text{sat}}(T_a) - m_a^v$ (see equations (52) and (53)), in the three models, System I (equations (40)–(43)), System A1 (equations (44)–(47)), and System A2 (equations (48)–(51)), described in the previous section, evaporation and condensation do not occur simultaneously. Owing also



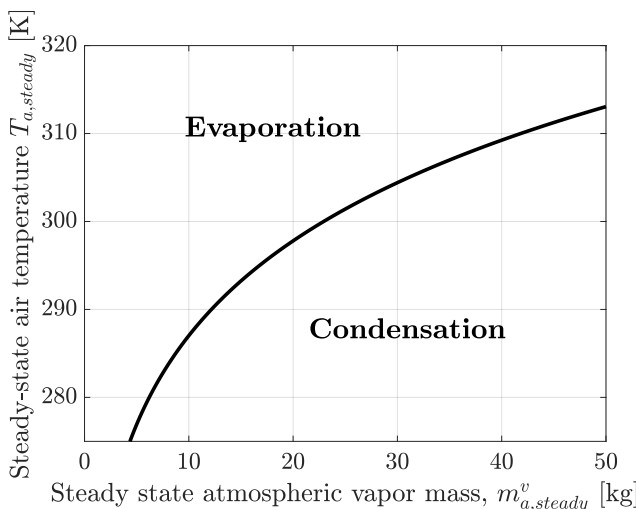

**Figure 2.** Steady-state air temperature $T_{a,\text{steady}}$ versus atmospheric water vapor mass, $m^v_{a,\text{steady}}$, computed from equation (55). The corresponding specific humidity $m^v_{a,\text{steady}}/(m^v_{a,\text{steady}} + m^d_a)$ for the range shown varies from 0 (dry) to 2% (humid).

to the opposite signs of the argument $((m^v_a + m^d_a)q_{\text{sat}}(T_a) - m^v_a)$ within the $\max$ functions in the definition of evaporation and condensation rates, the steady-state condition for vapor-liquid mass exchange in all models is given by $dV/dt = dK/dt = 0$. These conditions of zero evaporation and condensation also lead to a steady state in the rates of change of air and ocean temperatures in the governing ordinary differential equations. This equilibrium, corresponding to a fixed point of the ODEs, yields the relation:

$$q_{\text{sat}}(T_{a,\text{steady}}) = \frac{m^v_{a,\text{steady}}}{m^d_a + m^v_{a,\text{steady}}} \quad \Rightarrow \quad T_{a,\text{steady}}(m^v_{a,\text{steady}}) = \left( \frac{1}{T_0} - \frac{R_v}{L_v} \log \left( \frac{p_0}{c_1 \cdot c_2} \cdot \frac{m^v_{a,\text{steady}}}{m^d_a + m^v_{a,\text{steady}}} \right) \right)^{-1}. \quad (55)$$

Figure 2 shows how the steady-state air temperature $T_{a,\text{steady}}$ varies with the atmospheric vapor mass $m^v_{a,\text{steady}}$ for a representative set of physical parameters. As expected from the logarithmic form of equation (55), the rate of increase of $T_{a,\text{steady}}$ diminishes with increasing $m^v_{a,\text{steady}}$. The region above the neutral curve in the figure corresponds to finite evaporation of oceanic liquid into atmospheric vapor, while the region below indicates condensation.

The dynamical system described by each of the three models, I, A1, and A2, is fundamentally three-dimensional, as the atmospheric vapor mass changes at a rate equal and opposite to that of the oceanic liquid. In the phase space defined by $m^v_a$ (or equivalently $m^l_o$), $T_a$, and $T_o$, the steady-state (or neutral) curve shown in Fig. 2 represents the set of equilibrium points where vapor-liquid exchange is balanced. Notably, this curve is independent of the ocean temperature $T_o$. Consequently,





the full three-dimensional steady-state manifold is a surface formed by extruding the neutral curve of Fig. 2 along the $T_o$ axis. However, as discussed below, not all points on this surface correspond to stable equilibria. Moreover, the stability characteristics and dynamical trajectories differ significantly across the three models within realistic regimes of $T_a$, $T_o$, and $m_a^v$.

All three eigenvalues of both the current and ideal models are negative along the curve, indicating asymptotic stability (not shown). However, the Jacobian matrix is non-normal, implying that the transient growth due to linear mechanisms can be significant, even though perturbations ultimately decay. As we demonstrate below, these transient amplifications can drive trajectories far from the neutral curve, well beyond the regime of linear validity. In such cases, the full nonlinearity of the governing ODEs governs the long-term dynamics. Due to this non-normality, we do not present a detailed eigenvalue analysis.

Instead, we explore the system's behavior geometrically by examining representative trajectories and comparing the the three models through their phase portraits.

Figure 3 shows the evolution of six trajectories for each of the Systems I, A1, and A2, projected onto the $T_a$–$m_a^v$ plane (left panel) and in the three-dimensional space $T_o$–$T_a$–$m_a^v$ (right panel). In the condensation regime (i.e., $T_a < T_{a,\text{steady}}$, below the neutral curve in the left panel), the blue trajectories illustrate that, starting from the initial conditions (black circles), the

atmospheric vapor mass decreases in all three models up to their equilibrium values (black diamonds). Conversely, in the evaporation regime (orange trajectories), the vapor mass increases.

As evident from the left panel, condensation is accompanied by an increase in air temperature in all models. The equilibrium $T_a$ in System I lies between those of A1 and A2, with System A1 exhibiting the smallest increase in $T_a$. Also, in condensation regime, the ocean temperature (blue curves in the right panel) remains largely unaffected across all models. However, model

differences become more pronounced in the evaporation regime. In System I, latent heat release causes $T_a$ to decrease towards the neutral curve, consistent with physical expectations. In System A2, $T_a$ remains unchanged during evaporation, as this system lacks an evaporation-driven $T_a$ tendency term (see equation (49)). In contrast, System A1 exhibits an unphysical increase in $T_a$ and fails to reach equilibrium within realistic atmospheric values.

The reduction in ocean temperature during evaporation is smallest for System I (right panel of Fig. 3). Figure 4 further

illustrates the A1 behavior without limiting the display to realistic ranges. From initial conditions above the neutral curve (black circles in the left panel), $m_a^v$ increases as expected, but $T_a$ rises to unphysical values—up to 650 K—before approaching equilibrium at extremely large vapor masses (exceeding the dry air mass $m_a^d = 1000$ kg). Meanwhile, $T_o$ drops to unrealistically low, even negative, values. This behavior results from the evaporation-driven $T_a$ tendency term in equation (46). Since $c_l \approx 4c_p^d$, typical atmospheric values of $T_o$ and $T_a$ cause this term to push the system away from equilibrium. The trajectory reverses

only under extreme conditions where $T_o \lesssim T_a/4$. The distinction in the evaporation regime across the three systems can be also observed in Fig. 5, where the phase flow at a fixed $T_o = 295$K points towards the neutral curve in Systems I and A2, but away from it in A1.

Treating System I as the baseline, Fig. 6 quantifies the equilibrium errors in $m_a^v$, $T_o$, and $T_a$ for System A2—an E3SM-like system that does not conserve energy. The contours show the percentage change in the equilibrium values of $m_a^v$, $T_a$, and $T_o$

in System A2 relative to System I, for a fixed $T_o = 295$K, as a function of the initial atmospheric vapor mass and temperature $(m_a^v, T_a)$ shown along the x- and y-axes, respectively.




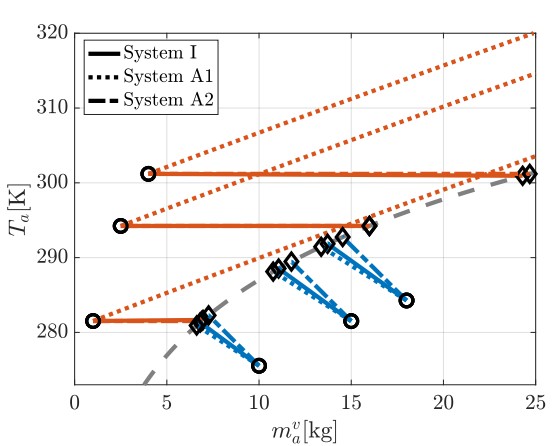
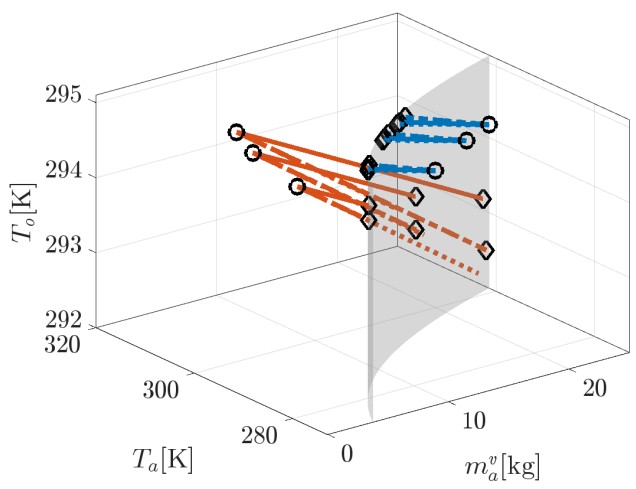

(a) 2D projection of trajectories in $T_a - m_a^v$ space
(b) Full 3D trajectories in $T_a - T_o - m_a^v$ space

**Figure 3.** Trajectories in the evaporation (orange) and condensation (blue) regimes predicted by the three models—System I (solid), A1 (dotted), and A2 (dash-dotted)—for realistic values of $T_a, T_o$ and $m_a^v$. **Left:** Projection onto the $T_a$–$m_a^v$ plane. **Right:** Full three-dimensional trajectories in $T_a - T_o - m_a^v$ space. The grey dashed curve (left) and the transparent surface (right) denote the steady-state (neutral) surface. Initial conditions are shown as black circles, and equilibrium points (when reached) as black diamonds.

Errors in vapor mass and air temperature are minimal near the neutral curve but grow in both condensation and evaporation regimes. The vapor mass error (left panel) is roughly ten times that of $T_a$ (middle panel), though both share similar spatial patterns. Ocean temperature is underpredicted by 3K (which is around $-1\%$) for high initial $T_a$.

Although System A1 conserves energy, its equilibrium errors mirror those in A2 but with significantly larger magnitudes. Figure 7 shows the logarithm of the absolute percentage errors in the equilibrium $m_a^v$, $T_a$, and $T_o$, computed for the same set of initial conditions used to visualize System A2's error in Fig. 6. In the condensation regime, errors are small (negative log contours). In contrast, the evaporation regime exhibits extreme errors: where the log of percentage error magnitude reaches 1 for $T_a$, 5 for $T_o$ and 10 for $m_a^v$. These substantial discrepancies reflect the destabilizing effect of A1's IEFLX evaporation term.


Finally, despite showing stability and smaller discrepancy relative to System I, A2 exhibits a net energy loss. Figure 8 (left) shows the percentage change in energy relative to its initial value (with $T_o = 295$ K). A loss of 1% is observed in evaporation,



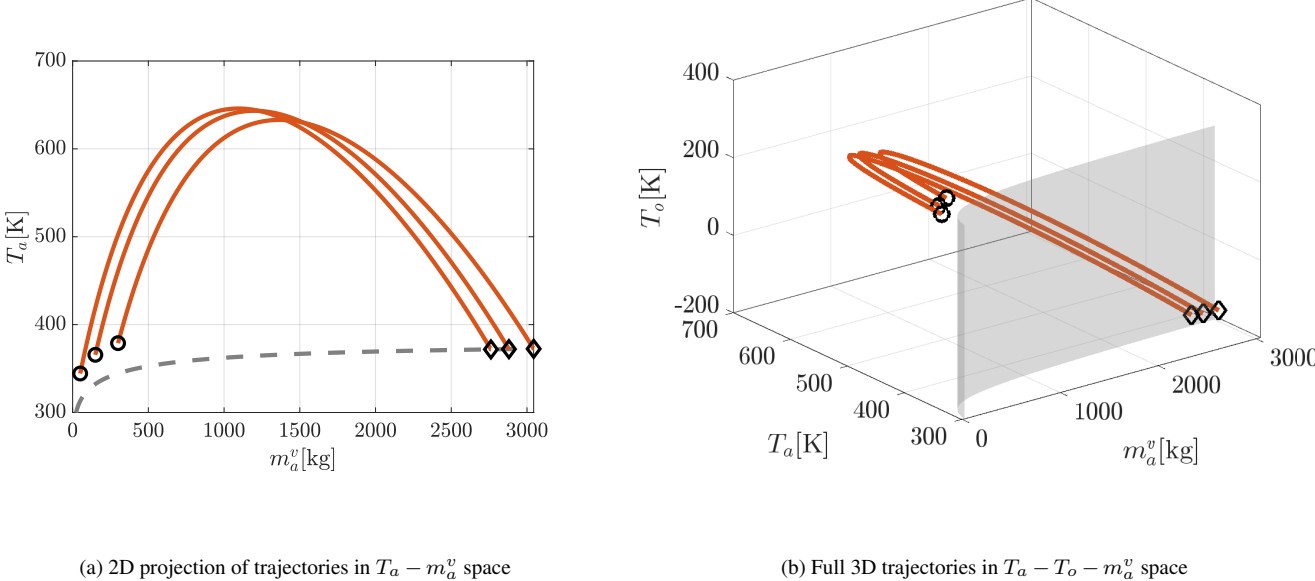

(a) 2D projection of trajectories in $T_a - m_a^v$ space $\qquad$ (b) Full 3D trajectories in $T_a - T_o - m_a^v$ space

**Figure 4.** Trajectories of System A1 in the evaporation regime up to the system equilibrium. Same symbols are used as in Fig. 3.

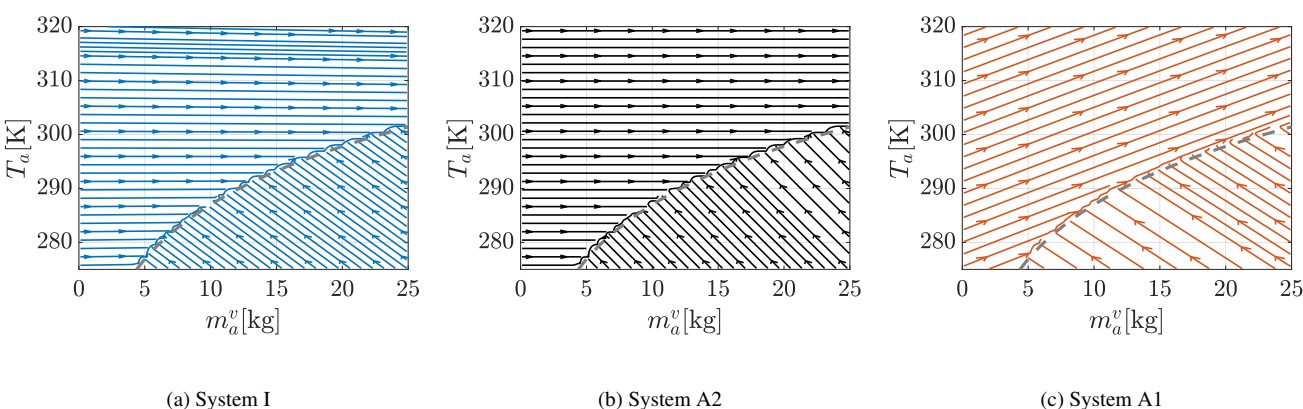

(a) System I $\qquad$ (b) System A2 $\qquad$ (c) System A1

**Figure 5.** Phase plots in $T_a - m_a^v$ plane of System (a) I, (b) A2 and (c) A1 at $T = 295$ K. Uneven streamline spacing is an artifact of MATLAB's streamslice function.





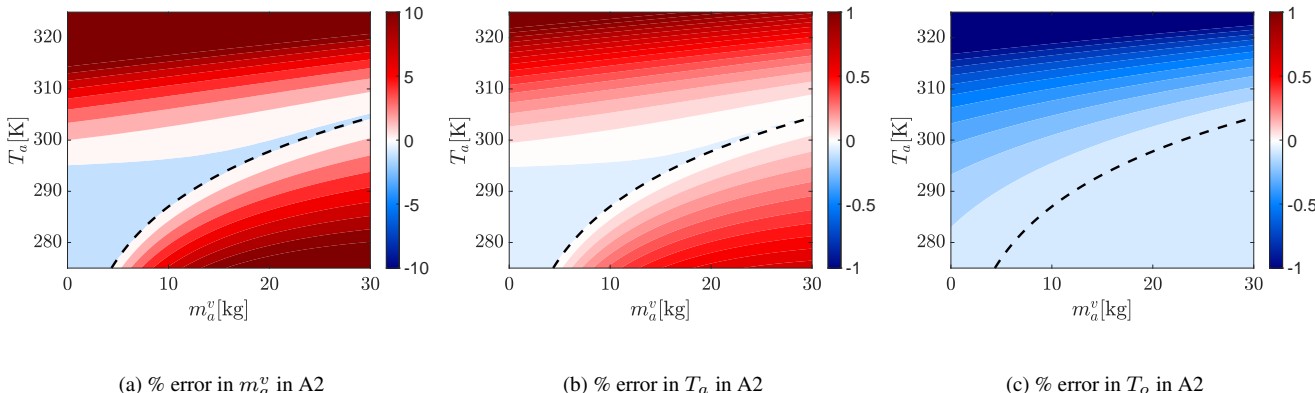

(a) % error in $m_a^v$ in A2      (b) % error in $T_a$ in A2      (c) % error in $T_o$ in A2

**Figure 6.** Percentage error in difference between equilibrium values of (a) $m_a^v$, (b) $T_a$ and (c) $T_o$ between Systems A2 and I for initial $T_o = 295$. The x and y axis correspond to the initial $T_a$ and $m_a^v$.

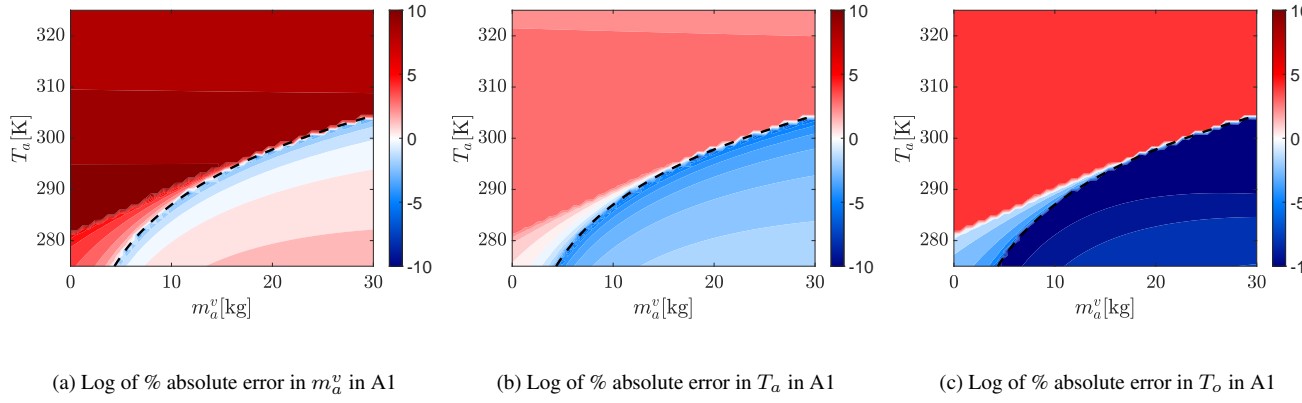

(a) Log of % absolute error in $m_a^v$ in A1      (b) Log of % absolute error in $T_a$ in A1      (c) Log of % absolute error in $T_o$ in A1

**Figure 7.** Logarithm of the absolute percentage error in equilibrium values of (a) $m_a^v$, (b) $T_a$ and (c) $T_o$ between Systems A1 and I for $T_o = 295$ K. Axes show initial $T_a$ and $m_a^v$.





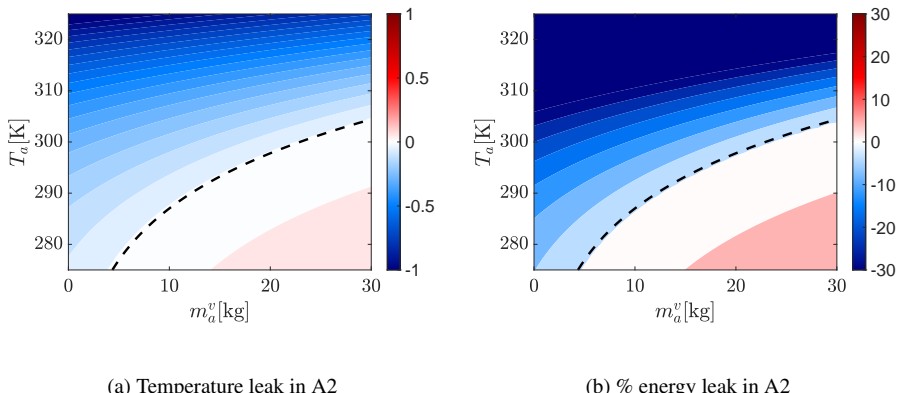

(a) Temperature leak in A2                                     (b) % energy leak in A2

**Figure 8.** (a) Temperature [K] and (b) percentage energy leak in A2 relative to the energy conserved by A1

with smaller losses in condensation. An alternate metric is the temperature leak, defined as

$$T_{\text{leak}} = \frac{E_{\text{final}} - E_{\text{initial}}}{c_p^d (m_a^d + m_{a,\text{final}}^v)}, \tag{56}$$

where, $E_{\text{final}} - E_{\text{initial}}$ is the energy lost and $m_{a,\text{final}}^v$ is the final mass of the water vapor in the atmosphere. Shown in the right
panel, it ranges from 0–30K in evaporation and is positive in condensation—qualitatively similar to the energy loss pattern.

## 5   Conclusions

We present a simplified framework to investigate moisture transfer mechanisms at the ocean-atmosphere interface for common
thermodynamic approximations and for *unapproximated* thermodynamics. The framework leads to systems of ODEs, which
represent E3SM and idealistic models. We study the models' behavior in evaporation and condensation regimes. Among all
of the models, only the idealistic one with unapproximated thermodynamics presents physically plausible behavior while
conserving energy.

*Code availability.* MATLAB scripts for all figures are located at https://doi.org/10.5281/zenodo.16858190 (Guba and Sharma, 2025, accessed August 13, 2025). Included README file contains instructions.

*Author contributions.* All authors contributed to conceptualization and manuscript writing. OG, AS, and MAT derived and implemented the
algorithms in MATLAB.



*Competing interests.* The authors declare that they have no conflict of interest.

*Acknowledgements.* Sandia National Laboratories is a multi-mission laboratory managed and operated by National Technology & Engineering Solutions of Sandia, LLC (NTESS), a wholly owned subsidiary of Honeywell International Inc., for the U.S. Department of Energy's
National Nuclear Security Administration (DOE/NNSA) under contract DE-NA0003525. This written work is authored by an employee of NTESS. The employee, not NTESS, owns the right, title and interest in and to the written work and is responsible for its contents. Any subjective views or opinions that might be expressed in the written work do not necessarily represent the views of the U.S. Government. The publisher acknowledges that the U.S. Government retains a non-exclusive, paid-up, irrevocable, world-wide license to publish or reproduce the published form of this written work or allow others to do so, for U.S. Government purposes. The DOE will provide public access to
results of federally sponsored research in accordance with the DOE Public Access Plan.

This work was supported by the Laboratory Directed Research and Development program at Sandia National Laboratories, a multimission laboratory managed and operated by National Technology and Engineering Solutions of Sandia LLC, a wholly owned subsidiary of Honeywell International Inc. for the U.S. Department of Energy's National Nuclear Security Administration under contract DE-NA0003525.

This research was supported as part of the Energy Exascale Earth System Model (E3SM) project, funded by the U.S. Department of
Energy (DOE), Office of Science, Office of Biological and Environmental Research (BER).



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
