# Peer review of "On moist ocean-atmosphere coupling mechanisms"

_EGUsphere, 2025_

## Referee Comment (RC1)

**On moist ocean-atmosphere coupling mechanisms**

The manuscript investigates sets of idealised equations used to describe the energy exchanges at an atmosphere-ocean interface in an Earth System model associated with evaporation of the ocean's liquid water into atmospheric vapor, and the condensation and subsequent precipitation of atmospheric vapor into the ocean. Three sets of equation are derived for a simple box model: System I derives some equations from "unapproximated" thermodynamics, System A1 attempts to emulate the behaviour of E3SM including energy fixers, while A2 makes the same thermodynamic simplifications as A1 but omits the energy fixers. Some characteristics of the solutions of these equation sets are then investigated numerically.

This paper is an attempt to understand a challenging and important issue in Earth System modelling: how to quantify the errors associated with thermodynamic simplifications made when coupling together different Earth System components, in the hope that this can lead to improvements in the modelled thermodynamics. The paper is well-written, so it is not too difficult to follow the mathematical derivations of the idealised equation sets, and I have no concerns about the quality of the work. The two most significant contributions of the paper are (a) the clear presentation of how evaporation from the ocean should be treated with unapproximated thermodynamics, and (b) the illustration of possible consequences of approximate treatments. My main suggestion is that the paper arrives at interesting results from their numerical experiments (particularly for System A1) but there isn't much discussion that links this back to expected behaviour in Earth System Models. For instance, the results suggest that System A1's evaporation process is unstable, so if possible it would be helpful to see discussion about how E3SM avoids this.

**1  Minor Comments**

There were a handful of points which I thought could benefit from clarification or expansion.

1. Page 4, Lines 103-105. The authors mention that for the processes relevant to this work, conservation of energy can be reduced to conservation of enthalpy. Although the authors provide citations for this, I think the paper would still benefit from a brief explanation of this.

2. Page 4, Line 106-109. Could the authors clarify that their $L_v$ and $L_l$ terms are constant, and so are the "reference" latent heats, rather than the standard latent heats (which should be temperature-dependent when the heat capacities of different phases of moisture are different).

3. Page 5, around Eqn (1): could you clarify that $m_o^l$ is the amount of water in the ocean *after* evaporation has occurred?

4. Page 5: Lines 141-143: Could the authors expand on why the atmosphere only receives an energy of $(L_v + L_l)\Delta m$, and why the pressure adjustment process being energy-conserving is the cause of this? Should this process not conserve energy?

5. Page 6, Line 157. I would remove the word "possibly", it seems clear to me that your framework is improved!

6. Page 6, Eqn (4). It might be helpful to clarify what the temperature is in this equation.

7. Page 6, Lines 172-179. I found this paragraph really useful in understanding Section 2.2, and wondered if the authors might consider moving some of it to be earlier, which might improve the readability of this section.

8. Page 7, Table 1: I found this table helpful. Can I just clarify whether the top left box ("energy flux received by atmosphere, current") should just be $(L_v + L_l)\Delta m$? Is it correct that it also includes $c_l T \Delta m$, as I felt this didn't agree with Line 143.

9. Page 12, Line 282-283: "as discussed later in Sec. 3.4.1". Apologies if I simply missed it, but I couldn't find this discussion.

10. Page 12, Eqn before Line 294. I had wondered here if the denominator should be $c_l(m_o^l + \Delta V)$?

11. Page 12, Eqn (25). I think it would be helpful in this equation to clarify that $m_l$ is $m_o^l$ to keep the notation consistent.

12. Page 19, Line 450, "In System I, latent heat release causes $T_a$ to decrease towards the neutral curve". I actually struggled to spot this decrease from the plot as the line seems almost flat, so it could be worth clarifying the description here.

13. Page 23, Conclusions: It would be really good to expand the conclusions, or the discussion in the previous section, to link the numerical results to the behaviour we might see from ESMs that follow the approaches of Systems A1 and A2, and why we might not see the instability that the numerical results seem to indicate. It would also be good to have a comment on how the authors intend to follow-up on the analysis from this work.

**2  Typographical or Style Points**

1. Page 3, Line 74: There is a space missing between "e.g.," and the citation

2. Page 4, Line 98: "boththe", space missing

3. Page 4, Line 104: "practive" should be "practice"

4. Page 5, Line 131: "Assume ocean" is missing "the"

5. Page 5, Lines 144 and 146, and Page 8 Line 218: there are forward apostrophes that should be backwards apostrophes.

6. Page 6, Line 178: is there a rogue space after the forward slash in "deficit/ excess"?

7. Page 17-18, Lines 415-419. This was a very long sentence! Could you break it up?